# Predicting the Unknown and the Unknowable. Are Anthropometric Measures and Fitness Profile Associated with the Outcome of a Simulated CrossFit^®^ Competition?

**DOI:** 10.3390/ijerph18073692

**Published:** 2021-04-01

**Authors:** Javier Peña, Daniel Moreno-Doutres, Iván Peña, Iván Chulvi-Medrano, Alberto Ortegón, Joan Aguilera-Castells, Bernat Buscà

**Affiliations:** 1Sport and Physical Activity Studies Centre (CEEAF), University of Vic—Central University of Catalonia, 08500 Vic, Spain; javier.pena@uvic.cat; 2Sport Performance Analysis Research Group (SPARG), University of Vic—Central University of Catalonia, 08500 Vic, Spain; 3ICON Training S.L., 08912 Barcelona, Spain; ivanpenalopez4@gmail.com; 4Sport Performance and Physical Fitness Research Group (UIRFIDE), Department of Physical and Sports Education, Faculty of Physical Activity and Sports Sciences, University of Valencia, 46010 Valencia, Spain; ivan.chulvi@uv.es; 5Department of Sports Sciences, Ramon Llull University, FPCEE Blanquerna, 08022 Barcelona, Spain; a.ortegon@outlook.es (A.O.); joanac1@blanquerna.url.edu (J.A.-C.); BernatBS@blanquerna.url.edu (B.B.)

**Keywords:** performance, athlete, high-intensity functional training, cross-training, functional fitness

## Abstract

The main objective of this research was to find associations between the outcome of a simulated CrossFit^®^ competition, anthropometric measures, and standardized fitness tests. Ten experienced male CrossFit^®^ athletes (age 28.8 ± 3.5 years; height 175 ± 10.0 cm; weight 80.3 ± 12.5 kg) participated in a simulated CrossFit^®^ competition with three benchmark workouts (“Fran”, “Isabel”, and “Kelly”) and underwent fitness tests. Participants were tested for anthropometric measures, sit and reach, squat jump (SJ), countermovement jump (CMJ), and Reactive Strength Index (RSI), and the load (LOAD) corresponding to the highest mean power value (POWER) in the snatch, bench press, and back squat exercises was determined using incremental tests. A bivariate correlation test and k-means cluster analysis to group individuals as either high-performance (HI) or low performance (LO) via Principal Component Analysis (PCA) were carried out. Pearson’s correlation coefficient two-tailed test showed that the only variable correlated with the final score was the snatch LOAD (*p* < 0.05). Six performance variables (SJ, CMJ, RSI, snatch LOAD, bench press LOAD, and back squat LOAD) explained 74.72% of the variance in a k = 2 means cluster model. When CrossFit^®^ performance groups HI and LO were compared to each other, *t*-test revealed no difference at a *p* ≤ 0.05 level. Snatch maximum power LOAD and the combination of six physical fitness tests partially explained the outcome of a simulated CrossFit competition. Coaches and practitioners can use these findings to achieve a better fit of the practices and workouts designed for their athletes.

## 1. Introduction

CrossFit^®^ is a training method property of CrossFit^®^ Inc. (Washington, DC, USA), a company established in 2000 by Greg Glassman and Laura Jenai. This form of physical exercise incorporates elements from other disciplines, such as weightlifting, powerlifting, gymnastics, calisthenics, and strength athletics, while following high-intensity exercise principles and using constant variability as one of its core elements. According to data from the company, the number of official CrossFit^®^ affiliated gyms in the world is close to 15,000 [1], a figure that shows the worldwide interest in this exercise regime. Apart from the CrossFit^®^ activity aimed at the general population, CrossFit^®^ Inc. has developed a competitive trend that also enjoys considerable international popularity. In 2019, 144,276 people completed all the workouts of the day (WODs) of the CrossFit Open^®^ as prescribed or “RX” [2] (meaning that the athletes used the prescriptive weight or height, completed the prescribed number of repetitions, and followed the full standards for each movement). Alongside 15 sanctioned events, the CrossFit Open^®^ is the only way to qualify for the CrossFit Games^®^, where the elite of this sport has convened every year since 2007.

Adult CrossFit^®^ participation seems to entail similar physical demands (in terms of VO_2_ max, muscle size, strength and endurance gains) to other high-intensity physical activities [3]. Several cohort studies have reported improvements in VO_2_ max [4,5], body composition [6,7], and specific work capacity [8] in men and women in interventions ranging from 6 to 10 weeks. Thus, CrossFit^®^ WODs are a demanding form of exercise, and physiologically, both aerobic and anaerobic metabolisms influence the athlete’s performance [9].

General strength improvements associated with CrossFit^®^ participation are also described in the literature with conflicting results. Significant increases in several muscular strength and endurance tests after participation in CrossFit^®^ workouts have been reported in some studies [5], while in some others, no significant differences were noted post-intervention [8].

However, all the studies mentioned above have two critical limitations highlighted in systematic reviews: a reduced number of scientific studies because the discipline is still incipient, and a lack of a high level of evidence at low risk of bias [10].

To date, several studies have highlighted that the physical stress caused by CrossFit^®^ WODs is comparable to a 20 min high-intensity treadmill run at 90% of maximal heart rate [11] and superior to an ACSM-based training session in terms of fatigue, muscle soreness, and muscle swelling [12]. Rating of perceived exertion (RPE) seems consistently high after CrossFit^®^ routines [12,13], and increased lactate [13,14,15] and pro/anti-inflammatory cytokine production [14] is also present in several scientific reports assessing these activities.

Although CrossFit^®^ athletic competitions generate significant revenues, not many previous studies have dealt with competitive performance factors. Numerous scientific contributions have investigated the epidemiology of CrossFit^®^ [3,16,17,18], with several cases of spinal injuries [19] and rhabdomyolysis [20] reported, but not many pieces of research have provided insight about the relevant elements of fitness to succeed in competitions. For instance, a study comparing the outcomes in three benchmark WODs—“Grace” (30 clean and jerks for time), “Fran” (three rounds of thrusters and pull-ups for 21, 15, and 9 repetitions), and “Cindy” (20 min of rounds of 5 pull-ups, 10 push-ups, and 15 bodyweight squats)—found that whole-body strength and anaerobic threshold exhibited association with specific CrossFit^®^ performance [21]. In a similar analysis with 32 healthy adult males, age, group (experienced vs. inexperienced), VO_2_ max, and anaerobic power were predictors of a 12 min as many repetitions as possible WOD with 12 throws of a 9.07 kg medicine ball at a 3.05 m target, 12 swings of a 16.38 kg kettlebell, and 12 burpee pull-ups [22]. In the same article, only CrossFit^®^ experience was a significant predictor in a WOD with sumo deadlift high pull, a 0.5 m box jump, and a 40 m farmer’s walk with 40 kg following a three-round with 21, 15, and 9 repetitions per exercise structure. Recent research has also found that absolute VO_2_ peak values and CrossFit^®^ Total (one repetition maximum tests for the back squat, deadlift, and overhead press) were predictors of the 19.1 CrossFit Open^®^ workout and the benchmark “Fran” performances, respectively [23]. Body composition was revealed as the most significant success predictor in the 2018 CrossFit Open^®^ [24].

Despite an increased number of scientific studies due to the growth in popularity of CrossFit^®^, there is still an important space for further research about CrossFit^®^ athletic competitions. The main objective of this cross-sectional study was to find associations between the outcome of a simulated CrossFit^®^ competition, anthropometric measures, and standardized fitness tests, providing insight to coaches and athletes to achieve better competitive performance.

## 2. Materials and Methods

### 2.1. Participants

A purposive sample of ten experienced male CrossFit^®^ athletes (age 28.8 ± 3.5 years; height 175 ± 10.0 cm; weight 80.3 ± 12.5 kg; one-hand reach 223 ± 15 cm) without relevant injuries at the moment of the study and recruited from official CrossFit^®^ affiliates volunteered to participate in the study. The inclusion criteria were set based on weekly training volume (≥5 sessions/week), competitive CrossFit^®^ background (≥2 years), regular participation in regional (*n* = 1), national (*n* = 5), or international (*n* = 4) competitions, and their ability to perform the RX versions of the workouts (respecting the metabolic purpose of the WOD and being able to lift the weights without fatal technical flaws in the presence of fatigue). Before starting the study, we informed the participants about the experimental procedures and they signed informed consent and provided additional data by filling out a modified Physical Activity Readiness Questionnaire (PAR-Q) [25]. Procedures followed the Declaration of Helsinki and its later amendments [26] and were approved by the Research Ethics Committee of the University of Vic - Central University of Catalonia in Barcelona, Spain (ref. no. 46/2018).

### 2.2. Experimental Procedures

Testing was conducted over two separate sessions. In the first session, before starting a simulated CrossFit^®^ competition, we tested the participants for anthropometric measures and a sit-and-reach flexibility test. Weight was assessed on an electronic scale (PS160, Beurer, Germany) with an accuracy of ±0.1 kg. Height was measured using a roll-up measuring tape with wall attachment (206, Seca^®^, Hamburg, Germany) with an accuracy of ±0.01 m. One-hand reach was assessed using a measuring tape (TM-CO2, Tacklife, New York, NY, USA). Body fat percentages were calculated using the equation of Jackson and Pollock [27] measuring the skinfold thickness at three sites (chest, abdomen, and thigh) using a caliper (Holtain Ltd. Tanner/Whitehouse Skinfold Caliper, Holtain, Dyfed, UK). One experienced anthropometrist carried out all the tests following the protocols established by the International Society for the Advancement of Kinanthropometry (ISAK). The sit-and-reach test was performed twice using a sit-and-reach box (Sit and Reach testing box, Eveque, Northwich, UK) and considering the best score as the final result in the test. Later, all of the participants completed three benchmark WODs in random order with a 30 min rest in between them, simulating a CrossFit^®^ Competition. The three selected WODs were “Fran”, “Isabel”, and “Kelly”, and they were performed in that same order (Table 1). These WODs were selected because they are popular benchmark WODs in the CrossFit^®^ community and because they incorporate very diverse skills and fitness elements (Olympic lifting movements, calisthenics, pure conditioning movements, and exercises with high VO_2_ max demands).

Every participant was assigned a certified CrossFit^®^ judge to control their performance, and the WODs were completed in two series or “heats”. Participants for the two heats in the first WOD were selected at random, while for the second and third WODs, the athletes with better accumulated scores were assigned to the second heat reproducing the usual CrossFit^®^ competition procedures. During the second session, a week later, we carried out the rest of the measurements (Table 2). Squat jump (SJ), countermovement jump (CMJ), and a 0.7 m drop jump (DJ) were measured using a contact mat (Ergojump-Plus, Ergotest Innovation, Norway) consisting of a switch mat connected to a digital timer (with an accuracy of ±0.001 s). Contact time and resulting height in the DJ were used to calculate Reactive Strength Index (RSI) by using the formula: RSI = Jump Height (cm)/Ground Contact Time (ms). All of the jumps were performed three times, and the best score was the final result in the tests. The loads (LOAD) corresponding to the highest mean power value (POWER) in the snatch, bench press, and back squat exercises were determined using incremental tests [28,29] and were measured with a linear encoder (MuscleLabTM, Ergotest Innovation, Stathelle, Norway) attached to the barbell. To assess the ability of the athletes to perform intermittent efforts, a Yo-Yo intermittent recovery test 2 (IR-2) was administered, and the distance covered was used to calculate VO_2_ max (mL/min/kg) using the formula: IR-2 distance (m) × 0.0136 + 45.3 [30]. All the mentioned tests were chosen because they show ecological and construct validity, the movements used are very similar to those of CrossFit^®^, and the tests enabling calculations have been validated by previous scientific literature.

### 2.3. Statistical Analysis

Using a statistical package (SPSS 21 for macOS, SPSS Inc, Chicago, IL, USA), a Shapiro–Wilk test was used to determine if the sample data was normally distributed prior to conducting a bivariate correlation test between the final competition score—assigning 10 points to the best-ranked competitor in each WOD, 9 to the next one, and consecutively so until the last competitor—and the different physical condition tests conducted in the study. Significance level was established at *p* < 0.05 (α = 5%) with a 95% confidence interval. In the second term, R, a language and environment for statistical computing (R 3.5.1 GUI 1.70 for macOS, R Foundation for Statistical Computing, Vienna, Austria), was used to normalize physical tests, centering them at 0 to avoid between-variable scale differences, carrying out a k-means cluster (k = 2) analysis considering the outcome of the physical tests to group individuals as either high (HI) or low (LO) performance. Later, a *t*-test was used to compare composite WOD scores between HI and LO groups. Finally, a Principal Component Analysis (PCA) was carried out to determine the influence of each physical test on the simulated CrossFit^®^ competition final composite score.

## 3. Results

The Shapiro–Wilk test showed that the variables included in the analysis were normally distributed (*p* > 0.05). A bivariate Pearson’s correlation coefficient two-tailed test of significance showed that the only variable showing a very large correlation [31] with the final score of the competition was the snatch LOAD (*p* < 0.05); none of the other variables showed association with the competition outcome (Table 3). Although weekly volume of training was not significantly correlated with the final competition score (*p* = 0.142), the r-value showed a promising correlation (0.50) with this factor.

A k-means model established two centroids that determined the two groups, HI (*n* = 6) and LO (*n* = 4) (Figure 1). The unpaired *t*-test comparison revealed no differences between HI and LO groups in WOD scores.

PCA cluster explains 74.72% of the variance using six performance variables measured in the study (SJ, CMJ, RSI, snatch LOAD, bench press LOAD, and back squat LOAD) (Figure 2). When CrossFit^®^ performance groups HI and LO were compared, the *t*-test revealed no difference at *p* ≤ 0.05 level.

The average values obtained in the tests included in the PCA are presented to describe the performances obtained by the athletes who participated in our study (Table 4).

## 4. Discussion

The purpose of this study was to determine if a battery of standardized physical fitness tests can predict the outcome of a simulated CrossFit^®^ competition. Competitive CrossFit^®^ is a complex discipline, where many different skills and elements of physical fitness (endurance, stamina, strength, flexibility, power, speed, coordination, agility, balance, and accuracy) come into play to achieve success. Due to this complexity, the CrossFit^®^ community has always accepted that the best way to assess performance (and therefore fitness levels) is to perform CrossFit^®^ benchmark WODs and participate in CrossFit^®^ competitions. This approach has significant limitations; specific CrossFit^®^ workouts test more than one capacity, making it difficult to attribute the progress in a workout to all of them equally. If we improve our time or repetitions in one particular CrossFit^®^ benchmark WOD, it is unfeasible to know if strength, skill, or conditioning was the main explanatory factor of this enhancement in performance. Additionally, CrossFit^®^ competitive performance requires psychological and physiological settings. Thus, understanding the attributes related with CrossFit^®^ performance can be relevant for two main reasons: it can be helpful to predict individual competitive outcomes and to work on the athletes’ weaknesses, improving their performances. 

Previous research has suggested a relationship between a combination of power measurements [22], whole-body strength [21], and power in the full-squat test [32], and CrossFit^®^ performance. However, this approach has limitations. On the one hand, it is undeniable that benchmarks and competitions are specific; they reproduce the “unknown and unknowable” axiom of the sport. Nevertheless, using them to test fitness can be time-consuming, and for some recreational athletes, the RX standards can be unachievable. In some WODs, this changes the “testing” conditions dramatically, because it is evident that it is not the same to perform the benchmark “Fran” with a 30 kg barbell and jumping pull-ups or to use the prescribed weight and movements in the RX version. Standardized tests are valid, reliable, accurate, and sensitive to detect changes in fitness, being useful in different populations and age groups. Their main disadvantage is the need for equipment that can be expensive and, in some cases, requires training to be used. However, their application is fast, and they equalize the execution conditions for everyone.

The data reported in the present study partially support the initial hypothesis. Only the result of one incremental test, the snatch, showed a strong (but not perfect) correlation with the outcome of the competition, and this was more than likely conditioned by the fact that one of the benchmark WODs in the event (“Isabel”) depended exclusively on the ability to perform this movement repeatedly with a high requirement of power. Despite this, the battery used in our study could discriminate between high (HI) and (low) LO performance athletes in the sample, explaining 74.72% of the variance with six performance variables measured. This result is consistent with that of other researchers arguing that CrossFit^®^ experience and training level is a critical component of performance in CrossFit^®^ workouts [22]. Weekly volume of training was not significantly correlated with the final competition score in our data, but a large correlation value (0.50) indicates that this factor can be considered as relevant in future research.

The lack of association between the individual outcome of the different fitness tests proposed and the simulated competition can be solved using a battery of tests. In one of the few investigations that we know regarding this matter, it was found that it is unfeasible to pretend that a single test of any nature can predict the result of a benchmark WOD in CrossFit^®^ [21]. 

Although the benchmark WODs in this study were selected because they present very different physical condition elements (aerobic and anaerobic demands, weightlifting, gymnastics, and conditioning movements), the variables that could explain the variance were all of a similar nature; the only test assessing VO_2_ max in our design showed no predictive power. “Fran” and “Isabel” are WODs that elite and sub-elite athletes can finish in less than five minutes, and “Kelly” lasts no longer than 20 min in these populations. This data agrees with previous research, where VO_2_ max did not predict CrossFit^®^ performance [21]. However, VO_2_ max has explained 68% of the variance in the outcome of the workout “Nancy” [33], with five rounds of 400 m run. In our case, the chosen workouts had an anaerobic predominance, and the rest periods between WODs were enough to emphasize the importance of muscular power in the competitive outcome, showing an enhanced specific work capacity in the athletes [8]. In this direction, a test using four consecutive Wingate anaerobic tests has predicted CrossFit^®^ specific performance in previous investigations [34].

We did not find any relationship between anthropometric measures and CrossFit^®^ specific performance. This may be attributed to the participants’ characteristics as expert athletes with suitable body composition (body fat 8.2 ± 2.83%) for their competitive development. The intrinsic characteristics of advanced CrossFit^®^ athletes and the purposive sampling used in this research piece may have been a limiting factor in finding an association between body composition and competition outcome. All the athletes in our sample clearly showed a physical condition above the average among CrossFit^®^ enthusiasts.

Flexibility levels were also shown not to be correlated with CrossFit^®^ performance in our study. To the best of our knowledge, no previous research has included flexibility as a possible predictor of CrossFit^®^ performance.

The present results should be interpreted with caution. The competition level of the athletes volunteering in our study (sub-elite) and the sample size are limitations to use our results to make inferences about other populations like elite athletes (CrossFit Games^®^ caliber) or inexperienced CrossFit^®^ recreational athletes. The selection of tests and benchmark WODs could also be a limitation. We should also understand that although all participants were instructed to perform all the WODs at the maximum intensity, the context (a simulated competition) could be less motivating than real competition settings. 

Future work on the current topic is therefore recommended to apply these findings to different cohorts, using other benchmark WODs or workouts from a real competitive event. Incorporating different standardized tests that can lead to more robust results, and a higher percentage of the variance of the outcome explained by the selected performance factors, could also be desirable.

This study set out to know in greater depth what the critical elements of physical fitness are that allow one to achieve a good result in a simulated CrossFit^®^ competition. The load at which the maximum snatch power was achieved and the combination of six physical fitness tests (SJ, CMJ, RSI, snatch LOAD, bench press LOAD, and back squat LOAD) partially explained the outcome of a simulated CrossFit^®^ competition with the benchmarks “Fran”, “Isabel”, and “Kelly”. Coaches and practitioners can use these findings to improve their decision-making processes and to use these tests as an element that can allow a better fit of the practices and workouts designed for their athletes.

## 5. Conclusions

Results coming from this article show that isolated physical condition tests can be misleading to explain the outcome of a CrossFit^®^ WOD. These individual tests can only be useful in cases where the benchmark WODs performed in the CrossFit^®^ context and its results are strongly related to the execution of one particular movement. Batteries of tests can help to discriminate athletes of different levels, showing that a better physical condition expressed in the battery is partially associated with a better overall performance in the specific CrossFit^®^ activity. These batteries should implement tests that are valid, reliable, accurate, and sensitive to detect changes in fitness, but at the same time show some level of specificity with competitive CrossFit^®^ requirements and CrossFit^®^ athletes’ specific needs.

## Figures and Tables

**Figure 1 ijerph-18-03692-f001:**
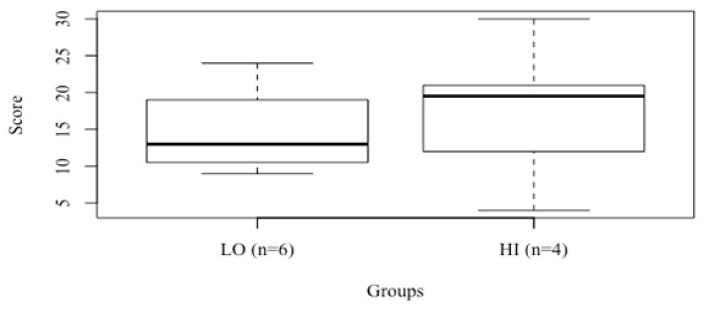
Boxplot visualization of the k-means cluster analysis grouping individuals as either high-performance (HI) or low performance (LO) and showing the minimum score, first quartile, median, third quartile, and maximum score achieved in the simulated competition by every group.

**Figure 2 ijerph-18-03692-f002:**
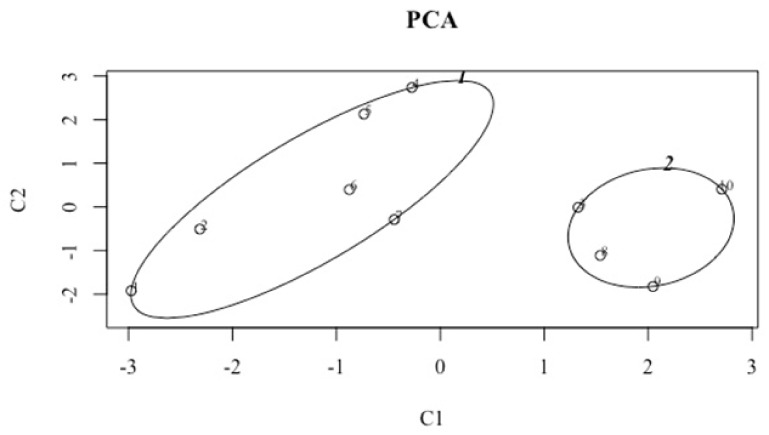
Principal Component Analysis with concentration and confidence ellipses around each group, including the six performance measures. Each main component is obtained by linear combination of the original six variables, and every dot inside the ellipses represents one individual in the HI (*n* = 6) and LO (*n* = 4) groups. These two components explain 74.72% of the point variability.

**Table 1 ijerph-18-03692-t001:** Workouts performed in the simulated CrossFit^®^ competition.

WOD 1 “FRAN”	WOD 2 “ISABEL”	WOD 3 “KELLY”
21-15-9 Repetitions of thrusters (42.5 kg) and pull-ups as fast as possible.	30 Repetitions of snatch (60 kg) as fast as possible.	Five rounds as fast as possible of 400 m run, 30 box jumps (0.5 meters), and 30 wall balls (9.07 kg medicine ball at a 3.05 m target).

**Table 2 ijerph-18-03692-t002:** Protocols followed in the incremental tests.

SNATCH	BENCH PRESS	BACK SQUAT
First load was set at the 65% of the one-repetition maximum (1RM) in the movement with 5% increments until failure.	Concentric execution of the exercise with 4 different loads ranging between 30 and 80% of the one-repetition maximum (1RM) in the movement.	Concentric execution of the exercise with 4 different loads ranging between 30 and 80% of the one-repetition maximum (1RM) in the movement.
Participants performed 2 repetitions at any given load with 10 s of rest between attempts and a 3 min rest between loads.	Participants performed 2 repetitions at any given load with 10 s of rest between attempts and a 3 min rest between loads.	Participants performed 2 repetitions at any given load with 10 s of rest between attempts and a 3 min rest between loads.

**Table 3 ijerph-18-03692-t003:** Correlation coefficients, interpretation, and significance levels in the variables included in the study.

Variables	Correlation (r and Interpretation)	Significance (*p*-Value)
Age (y)	−0.36, moderate	0.300
Weight (kg)	0.12, small	0.736
Height (cm)	0.25, small	0.490
Reach (cm)	0.21, small	0.566
Hours of training per week (h)	0.50, large	0.142
Body fat %	0.06, trivial	0.874
Sit and reach (cm)	0.05, trivial	0.896
Squat jumpJ (cm)	0.27, small	0.452
Countermovement jump (cm)	0.31, medium	0.390
Reactive strength index	0.14, small	0.695
Snatch LOAD (kg)	0.74, very large	0.014 *
Snatch POWER (W)	−0.13, small	0.721
Bench press LOAD (kg)	0.32, moderate	0.368
Bench press POWER (W)	0.34, moderate	0.337
Back squat LOAD (kg)	0.30, moderate	0.392
Back squat POWER (W)	0.2, trivial	0.548
Yo-Yo test IR-2 (m)	0.40, moderate	0.253

* Denotes significant correlation (*p* < 0.05).

**Table 4 ijerph-18-03692-t004:** Average values obtained in the tests included in the PCA.

Descriptive Statistics	SJ (cm)	CMJ (cm)	RSI	Snatch LOAD (kg)	Bench Press LOAD (kg)	Squat LOAD (kg)
Mean	33.1	38.1	0.114	59.6	53.8	65.7
Standard deviation	8.7	7.2	0.033	9.7	14.8	21.6

## Data Availability

The data that support the findings of this study are available on request from the corresponding author.

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
