# Peer review of "Predicting the Unknown and the Unknowable. Are Anthropometric Measures and Fitness Profile Associated with the Outcome of a Simulated CrossFit® Competition?"

_ijerph, 2021, doi:10.3390/ijerph18073692_

Round 1

Reviewer 1 Report

In general, this could be a potentially important paper but it could use some  improvements for example: (1) adding all abbreviated words into the keyword section with abbreviation as a quick reference, (2) aligning column headings on Table 3, (3) consider revising Figures 1 and 2 as they are hard to decipher.  

Author Response

Rebuttal letter

Please note that:

R_: reviewer comment

AR: authors response

REVIEWER #1

R1: In general, this could be a potentially important paper but it could use some improvements for example: (1) adding all abbreviated words into the keyword section with abbreviation as a quick reference, (2) aligning column headings on Table 3, (3) consider revising Figures 1 and 2 as they are hard to decipher. 

AR: Thanks for your words. We have addressed your comments separately below.

  • When it comes to adding the abbreviated words into the keywords section, although IJERPH accepts free-format submissions, we have been unable to detect any other paper published in the journal incorporating the abbreviations in that section. For the sake of homogeneity, we have decided to maintain the current structure of the paper.
  • The headings in Table 3 have been corrected and aligned properly. Thanks for the comment.
  • To improve the understanding of the figures we have provided more information in the captions. Now, it can be read as follows:

“Figure 1. Boxplot visualization of the K-means cluster analysis grouping individuals as either high-performance (HI) or low performance (LO) and showing the minimum score, first quartile, median, third quartile, and the maximum score achieved in the simulated competition by every group”.

“Figure 2. Principal Component Analysis with concentration and confidence ellipses around each group, including the six performance measures. Each main component is obtained by linear combination of the original six variables, and every dot inside the ellipses represents one individual in the HI (n=6) and LO (n=4) groups. These two components explain 74.72% of the point variability”.

Reviewer 2 Report

The objective of the authors was to find associations between the result of a simulated CrossFit® competition, anthropometric measurements and standardized fitness tests. Overall, the authors achieved the goal and the new information will help coaches and practitioners improve their decision on the best batteries, increasing athletes' performance.

The written is clearly and well based on literature. I only suggest to the authors to change the tables format because the informations are confusing.

1)Table 2, in the Bench press column the word ranging is together with the word exercise of the Back Squat column;

2) Table 3 is not formatted and the title of each column is in the wrong place.

Author Response

REVIEWER #2

R2: The objective of the authors was to find associations between the result of a simulated CrossFit® competition, anthropometric measurements and standardized fitness tests. Overall, the authors achieved the goal and the new information will help coaches and practitioners improve their decision on the best batteries, increasing athletes' performance.

AR: Thanks for your opinion on the paper. We are glad that the reviewer considers that we have achieved our main goal.

R2: The written is clearly and well based on literature. I only suggest to the authors to change the tables format because the informations are confusing.

  • Table 2, in the Bench press column the word ranging is together with the word exercise of the Back Squat column.

AR: Thanks for the comment, we have reformatted Table 2 to provide a better understanding of the information contained

R2: 2) Table 3 is not formatted and the title of each column is in the wrong place.

AR: Thanks for spotting that flaw in Table 3. As we have commented to the first reviewer, the headings in Table 3 have been corrected and aligned properly.

Reviewer 3 Report

The paper deals with an interesting research with implications for coaches, practitioners, and athletes for better design of the workouts. The work is methodologically correct and the paper fits within the scope of the journal.

However, I consider the authors should improve their work in some way.

General comments

  1. The aim of the paper could be clearer in the content of the manuscript [page 3, lines 101-103]. Please, review the aim and consider rewriting it in a clearer way, similar to the abstract. I believe it would help to a better understanding by the potential reader.
  2. The issue that worries me the most is related to the sample. To what extent can we extrapolate the results of your research in other contexts? Considering that the sample size is small (10 athletes), and all of them, are male, I feel this can be an important limitation of the study. What is the reason for focusing on only 12 male players? Do the authors feel that this could be a methodological flaw in the study?
  3. In this line, I consider the authors could better explain the characteristics of the sample. For example, it would be necessary to include in the manuscript the inclusion/exclusion criteria.

Specific comments

[Page 2, lines 46-48] Could the authors provide some evidence of the following?: “According to data from the 46 company, the number of official CrossFit® affiliated gyms in the world is close to 14,000, 47 a figure that shows the worldwide interest in this exercise regime”. It is advisable to add a reference.

[Page 2, lines 50-55] Could the authors provide references about the ideas in these lines?

[Page 2, lines 97-98] Please review the sentence “Body composition revealed AS A THE most significant success predictor in the 2018 CrossFit Open®”. Check that the intended meaning is retained and rewrite it if applicable.

Minor comments

[Table 3] Please review the table format: first row.

[Figures] According to the journal guidelines, all figures should be cited in the main text as Figure 1, Figure 2 (not as Fig 1, Fig 2).

[Page 7, line 300] “This study set out to know” should be corrected by “This study sets out to know”.

The effort made by the authors is appreciated and the comments are expected to be useful.

Author Response

REVIEWER #3

R3: The paper deals with an interesting research with implications for coaches, practitioners, and athletes for better design of the workouts. The work is methodologically correct and the paper fits within the scope of the journal. However, I consider the authors should improve their work in some way.

AR: Thanks for your comments about the manuscript. We will try to address all your comments and concerns to improve the quality of the article.

R3: The aim of the paper could be clearer in the content of the manuscript [page 3, lines 101-103]. Please, review the aim and consider rewriting it in a clearer way, similar to the abstract. I believe it would help to a better understanding by the potential reader.

AR: We have rephrased the paragraph mentioned above as follows: “The main objective of this cross-sectional study was to find associations between the outcome of a simulated CrossFit® competition, anthropometric measures and standardized fitness tests, providing insight to coaches and athletes to achieve better competitive performance”.

R3: The issue that worries me the most is related to the sample. To what extent can we extrapolate the results of your research in other contexts? Considering that the sample size is small (10 athletes), and all of them, are male, I feel this can be an important limitation of the study. What is the reason for focusing on only 12 male players? Do the authors feel that this could be a methodological flaw in the study?

AR: Thanks. Although common in sports science studies, small samples are always an element that limits the generalizability of the results found. In our study, we do not say at any point that our results are generalizable. We want to show the CrossFit community that there are alternative ways to assess athletes' preparation in this modality. However, to minimize the reviewer's concern regarding this aspect, we have included the following text in the limitations passages of the discussion: “The present results should be interpreted with caution. The competition level of the athletes volunteering in our study (sub-elite) and the sample size are limitations to infer our results to other populations like elite athletes (CrossFit Games® caliber) or inexperienced CrossFit® recreational athletes”.

R3: In this line, I consider the authors could better explain the characteristics of the sample. For example, it would be necessary to include in the manuscript the inclusion/exclusion criteria.

AR: Thanks for your comment, we have modified some passages of the participants section to meet the reviewer’s concerns. Now it can be read in the manuscript as follows: “A purposive sample of ten experienced male CrossFit® athletes (age 28.8 ± 3.5 years; height 175 ± 10.0 cm; weight 80.3 ± 12.5 kg; one-hand reach 223 ±15 cm) without relevant injuries at the moment of the study and recruited from official CrossFit® affiliates volunteered to participate in the study. The inclusion criteria were set based on weekly training volume (≥5 sessions/week), competitive CrossFit® background (≥2 years), regular participation in regional (n = 1), national (n = 5) or international (n = 4) competitions, and their ability to perform the RX versions of the workouts (respecting the metabolic purpose of the WOD and being able to lift the weights without fatal technical flaws in the presence of fatigue)”.

R3: [Page 2, lines 46-48] Could the authors provide some evidence of the following?: “According to data from the 46 company, the number of official CrossFit® affiliated gyms in the world is close to 14,000, 47 a figure that shows the worldwide interest in this exercise regime”. It is advisable to add a reference.

R3: [Page 2, lines 50-55] Could the authors provide references about the ideas in these lines?

AR: We have provided two references to back-up our statements. Additionally, the figure of CrossFit® affiliated gyms in the world has been updated.

R3: [Page 2, lines 97-98] Please review the sentence “Body composition revealed AS A THE most significant success predictor in the 2018 CrossFit Open®”. Check that the intended meaning is retained and rewrite it if applicable.

AR: Thanks for pointing out the typo. It has been corrected.

R3: Please review the table format: first row.

AR: The error was noticed by previous reviewers and it has been amended. Thanks

R3: [Figures] According to the journal guidelines, all figures should be cited in the main text as Figure 1, Figure 2 (not as Fig 1, Fig 2).

AR: Thank you for making us notice this oversight. It has been corrected

R3: [Page 7, line 300] “This study set out to know” should be corrected by “This study sets out to know”.

AR: Thanks for the comment. We have corrected the tense and person.

Reviewer 4 Report

This study analyses the associations between anthropometric measures and standardized fitness tests with the result of a simulated CrossFit® competition of ten experienced male CrossFit® athletes. The only variable correlated with the final score of the simulated competition was the Snatch LOAD and six tests explained 74.72% of the variance between high and low performance groups.

In my opinion, it is good research work that will be a reference for future studies on this field.

Specific comments

Since all the subjects in the sample frequently compete in national and international competitions, why didn't you analyze the results obtained in real competitions instead of mock competitions? Please, justify.

Line 113. The number of athletes participating in national (n = 5) and international (n = 4) (TOTAL n = 9) competitions does not correspond to the number of the sample (n = 10).

Line 113-114. It must be explained how the ability to perform the RX versions of the training was evaluated.

Lines 136-138 If WODs were included to incorporate very diverse abilities and fitness elements, a better description of the exercises and physical fitness requirements must be explained in greater depth.

Line 150-157. A greater justification is needed about why these tests and protocols were selected.

Line 152 Correct ".7"

Line 164 Table 2. Please, increase the column margins and text alignment.

Line 174-176. If the sample consisted of expert athletes, with similar characteristics in terms of training volume, competitive experience, participation in competitions, etc. It does not seem appropriate to group them into two clusters where one of them is “Low performance”. In addition, more information should be included regarding the results of the tests obtained by both groups.

Line 184. "(significance established at p <.05 with a 95% confidence interval)". This information should be included in the statistical analysis.

Line 189. Table 3. Please, check the format.

DISCUSSION

Line 281-284. This should be stated as a limitation of the study. Having selected such a specific study sample, it is difficult to find a relationship between anthropometric characteristics and performance in competition.

Author Response

REVIEWER #4

R4: This study analyses the associations between anthropometric measures and standardized fitness tests with the result of a simulated CrossFit® competition of ten experienced male CrossFit® athletes. The only variable correlated with the final score of the simulated competition was the Snatch LOAD and six tests explained 74.72% of the variance between high and low performance groups. In my opinion, it is good research work that will be a reference for future studies on this field.

AR: Thanks for your words about the paper and its potential interest.

R4: Since all the subjects in the sample frequently compete in national and international competitions, why didn't you analyze the results obtained in real competitions instead of mock competitions? Please, justify.

AR: Thanks for your question. A study with similar characteristics to the present study within a competitive setting is one of our future research lines. However, we initially ruled it out for various reasons. In the first place, logistical issues (access to the facilities, organizational permits); on the other hand, many confounding factors can alter the observations (the WODs are not carried out at the same time, the athletes do not have all the same rest time between WOD's as they are organized in heats according to their score). Additionally, in a top-level CrossFit competition, participating athletes have different geographical origins. This fact limits the follow-up, and as we describe in our methodology, our study needed two sessions with a washout period to be completed.

R4: Line 113. The number of athletes participating in national (n = 5) and international (n = 4) (TOTAL n = 9) competitions does not correspond to the number of the sample (n = 10).

AR: Thanks for pointing out that aspect. In fact, there is no error on these figures. One of the participants met the inclusion criteria but had no previous experience in national or international competitions. To highlight the quality of the sample this data was omitted, but we agree with the reviewer that should appear. Now this section can be read as follows: “The inclusion criteria were set based on weekly training volume (≥5 sessions/week), competitive CrossFit® background (≥2 years), regular participation in regional (n = 1), national (n = 5) or international (n = 4) competitions, and their ability to perform the RX versions of the workouts”.

R4: Line 113-114. It must be explained how the ability to perform the RX versions of the training was evaluated.

AR: All the athletes in the sample had prior experience with the selected WODs and had performed them on various occasions before the investigation. The WODs selected for the simulated competition are well known in the CrossFit world, and participants were purposively recruited with these prerequisites. However, we have performed a clarification in this regard in the text that we hope will help understand that the athletes met these standards. Thanks. “and their ability to perform the RX versions of the workouts (respecting the metabolic purpose of the WOD and being able to lift the weights without fatal technical flaws in the presence of fatigue)”.

R4: Lines 136-138 If WODs were included to incorporate very diverse abilities and fitness elements, a better description of the exercises and physical fitness requirements must be explained in greater depth.

AR: Thanks for the comments. Further explanation about the requirements of the WODs has been added. The paragraph can now be read as follows: “These WODs were selected for being popular benchmark WODs in the CrossFit® community and for incorporating very diverse skills and fitness elements (Olympic lifting movements, calisthenics, pure conditioning movements, and exercises with high VO2 max demands)”.

R4: Line 150-157. A greater justification is needed about why these tests and protocols were selected.

AR: The following information has been added to clarify the selection of these tests: “All the mentioned tests were chosen because they show ecological and construct validity, the movements used are very similar to those of CrossFit®, and the tests enabling calculations have been validated by previous scientific literature”. Thanks.

R4: Line 152 Correct ".7"

AR: Thanks. Amended.

R4: Line 164 Table 2. Please, increase the column margins and text alignment.

AR: Corrected, thanks.

R4: Line 174-176. If the sample consisted of expert athletes, with similar characteristics in terms of training volume, competitive experience, participation in competitions, etc. It does not seem appropriate to group them into two clusters where one of them is “Low performance”. In addition, more information should be included regarding the results of the tests obtained by both groups.

AR: In response to the reviewer's comments, we have added some information to lines 265-294 ("Statistical analysis") that we hope can clarify the concerns. As stated in the manuscript, the only variable associated with the simulated competition's outcome was the Snatch LOAD (p < .05). The PCA determined that some of the performed tests (SJ, CMJ, RSI, snatch LOAD, bench press LOAD, and back squat LOAD) explained 74.72% of the variance, and the unpaired t-test revealed no differences between the groups determined by the k-means cluster at a p ≤ .05 level. The clustering modelling allowed us to advance our conclusions further and not base our results solely on the association found. However, as we commented on several occasions in the article, these tests only explain the competitive outcome partially, being one of the limitations of our study. To clarify the rationale of the statistical tests used, the division between participants' levels was not set aprioristically. All the participants had a similar profile and met the inclusion criteria. However, a posteriori, the selected physical tests were related to the specific CrossFit® performance (composite WOD score). The k-means established the two performance groups and enabled the PCA to determine that some tests were promising, explaining a significant percentage of the variance observed. Also, to enhance this understanding and following the reviewer suggestion, a table showing descriptive statistics (mean and SD) of the values obtained by the athletes participating in our study in each of the tests included in the PCA has been added to the manuscript.

R4: Line 184. "(significance established at p <.05 with a 95% confidence interval)". This information should be included in the statistical analysis.

AR: The information has been added to the statistical analyses as follows: “Significance level was established at p < .05 (α = 5%) with a 95% Confidence Interval”.

R4: Line 189. Table 3. Please, check the format.

AR: Thanks, Amended.

R4: Line 281-284. This should be stated as a limitation of the study. Having selected such a specific study sample, it is difficult to find a relationship between anthropometric characteristics and performance in competition.

AR: Thanks for expressing your concern on the matter. We have added the following statement at the end of that same paragraph: “The intrinsic characteristics of advanced CrossFit® athletes and the purposive sampling used in this research piece may have been a limiting factor in finding an association between body composition and competition outcome. All the athletes in our sample clearly showed a physical condition above the average among CrossFit® enthusiasts”.

Round 2

Reviewer 3 Report

The authors have nicely addressed all the concerns raised in the previous round. Now I feel the paper is more consistent and has more relevance for the potential reader. Congratulations on a good job.